

# Marine monitoring in Europe: is it adequate to address environmental threats and pressures?

Suzanne J. Painting[1], Kate A Collingridge[1], Dominique Durand[4], Antoine Grémare[2], Veronique Créach[1], Christos Arvanitidis[3], Guillaume Bernard[5]

Affiliations:

[1] Centre for Environment, Fisheries and Aquaculture Science (CEFAS), Lowestoft, NR33 0HT, UK.

[2] Université de Bordeaux, EPOC, UMR 5805, F-33615 Pessac, France

[3] Hellenic Centre for Marine Research, Heraklion, 71500, Greece

[4] COVARTEC, Bergen, 5141, Norway.

[5] CNRS, EPOC, UMR 5805, F-33615 Pessac, France

*Correspondence to*: Suzanne J. Painting (suzanne.painting@cefas.co.uk)



**Abstract**
We provide a review of the environmental threats and gaps in monitoring programmes in
European coastal waters based on previous studies, an online questionnaire, and an in-depth
assessment of observation scales. Our findings underpin the JERICO-NEXT[1] monitoring
strategy for the development and integration of coastal observatories in Europe, and support
JERICO-RI[2] in providing high-value physical, chemical and biological datasets for addressing
key challenges at a European level. This study highlights the need for improved monitoring of
environmental threats in European coastal environments.
Participants in the online questionnaire provided new insights into gaps between environmental
threats and monitoring of impacts. In total, 36 national representatives, scientists and
monitoring authorities from 12 European countries (Finland, France, Germany, Greece,
Ireland, Italy, Malta, Norway, Poland, Spain, Sweden, United Kingdom) completed the
questionnaire, and 38 monitoring programmes were reported. The main policy drivers of
monitoring were identified as the EU Water Framework Directive (WFD), Marine Strategy
Framework Directive (MSFD), Regional Seas conventions (e.g. OSPAR) and local drivers.
Although policy drivers change over time, their overall purposes remain similar. The most
commonly identified threats to the marine environment were: marine litter, shipping,
contaminants, organic enrichment, and fishing. Regime shift was identified as a pressure by
67% of respondents. The main impacts of these pressures or threats were identified by the
majority of respondents (>70%) to be habitat loss or destruction, underwater noise, and
contamination, with 60% identifying undesirable disturbance (e.g. oxygen depletion), changes
in sediment/substrate composition, changes in community composition, harmful micro-
organisms and invasive species as key impacts.
Most respondents considered current monitoring of threats to be partially adequate or not
adequate. The majority of responses were related to spatial and/or temporal scales at which
monitoring takes place, and inadequate monitoring of particular parameters. Suggestions for
improved monitoring programmes included improved design, increased monitoring effort and
better linkages with research and new technologies. Improved monitoring programmes should
be fit-for-purpose, underpin longer-term scientific objectives which cut across policy and other
drivers, and consider cumulative effects of multiple pressures.
The JERICO-RI aims to fill some of the observation gaps in monitoring programmes through
development of new technologies. The science strategy for JERICO-RI will pave the way to a
better integration of physical, chemical and biological observations into an ecological process
perspective.

---

[1] JERICO-NEXT is the European H2020 project under grant agreement No. 654410.

[2] JERICO-RI is the European coastal research infrastructure (RI) community built by and through JERICO-
NEXT and its predecessor JERICO (Framework 7 Grant Agreement 49 no 262584).





## 1. Introduction

Across the globe, marine monitoring networks are becoming increasingly important for the collection, dissemination and sharing of data for improved scientific understanding, assessment of the health of marine ecosystems and forecasting the likely impacts of environmental change and human activities (Schofield et al 2002; Schofield et al 2003; Proctor and Howarth 2008; Duarte et al 2018; Buck et al 2019; Grand et al 2019; Smith et al 2019a; Smith et al 2019b). In Europe, for example, projects and infrastructures such as JERICO [3], DEVOTES [4], COPERNICUS[5], EMODnet[6], EMSO ERIC[7], and AtlantOS[8] have played a significant role in the co-ordination and advancement of monitoring in coastal and offshore waters, from operational marine services through to delivering data products to end users. Changing pressures (e.g. due to population growth and climate change) and changing requirements to monitor, manage and mitigate the impacts of pressures require ongoing review of monitoring programmes. Over the past few decades, marine monitoring has been implemented in coastal and shelf seas around Europe in response to local/regional monitoring and oceanographic research demands. However, heterogeneity in monitoring methods and approaches has limited the integration of coastal observations. Many of the observations are driven by short-term research projects, potentially limiting the sustainability of observing systems for meeting monitoring and assessment needs.

The Dobris Assessment (EEA 1995) listed 56 broad environmental threats, 19 of which were relevant to the coastal domain. These include physical modifications (e.g. due to urban development, industry, energy production, military activities, fisheries, recreation), contamination and coastal pollution (e.g. due to wastewater disposal, chemical contaminants, marine litter) and loss of biodiversity and genetic resources. Recent EU policy drivers and regional sea conventions have led to improvements in water quality in many regions (notably the Baltic Sea, North Sea, Celtic Sea, Bay of Biscay). Nonetheless, the fourth assessment of the European environment (EEA 2008a; see also EEA 2015a) highlighted that some regions remain affected by eutrophication, destructive fishing practices, hazardous substances, oil pollution and invasive species. Key concerns include increasing population densities and development of built-up areas, and likely impacts of climate change on physical (e.g.

---

[3] http://www.jerico-ri.eu/previous-project/jerico-fp7/
[4] http://www.devotes-project.eu/
[5] https://www.copernicus.eu/en
[6] http://www.emodnet.eu/
[7] http://www.emso.eu/
[8] https://www.atlantos-h2020.eu/



temperature, currents), chemical (e.g. acidification) and biological (e.g. changes in growth,
species composition and distribution, loss of organisms with carbonate shells) components.
The lack of comparable data presents a major obstacle for assessments of Europe's regional
seas, even for well-known problems such as eutrophication (EEA 2008b; OSPAR 2017). More
and better data are needed to develop a pan-European marine protection framework that
addresses environmental issues in a cost-effective way.
A number of studies have considered the suitability of monitoring programmes in Europe (e.g.
Bean et al 2017; Borja et al 2019; DEVOTES[9]; Garcia-Garcia et al 2019; Tett et al 2013;
Zampoukos et al 2013) for assessing good environmental status (GES) of the biodiversity suite
of MSFD descriptors (D): D1 (biodiversity), D2 (non-indigenous species), D4 (food-webs) and
D6 (seafloor integrity). Limitations have been identified in monitoring programmes, including
limitations in spatial and temporal coverage, pressures addressed, integrated monitoring
(addressing more than one descriptor and/or ecosystem component simultaneously), indicators
used, and data accessibility. Differences between countries highlight budgetary constraints and
differing approaches to monitoring. The Baltic region has been shown to be good at addressing
multiple descriptors simultaneously, while the Mediterranean has a good history of co-
ordination between countries and making good use of citizen science. Improved compatibility
of datasets (for example, through standardisation of sampling methods and quality assurance
of the data) and translating research activities into monitoring (e.g. for litter and noise) have
been highlighted as key challenges (EEA 2008a; EU DEVOTES).
The current EU JERICO-NEXT[10] project addresses the challenges of observing the complex
and highly variable coastal seas at a Pan-European level, in order to improve operational marine
services and meet the requirements of key policy drivers such as EU Directives. The emphasis
in JERICO-NEXT is on providing an integrated European observing system supporting
improved understanding of the coupling between physics, biogeochemistry and biology to take
account of and address the complexity of the coastal environment. This requires development
and application of new technologies that allow for the continuous monitoring of a larger set of
parameters. It also requires an *a priori* definition of the optimal sampling strategy over very
different spatial and temporal scales to develop fit-for-purpose coherent monitoring
programmes. This will enable JERICO-NEXT to meet the overall objective of extending the

---

[9] DEVOTES is an EU FP7 project

[10] JERICO-NEXT is a Horizon 2020 funded project, implementing the second phase of the European JERICO-RI research infrastructure aiming at multidisciplinary observations of coastal and shelf seas.





EU network of coastal observations developed during JERICO-FP7. As part of the JERICO-
NEXT project, we conducted an opinion poll of experts in European countries (Figure 1) to
identify current and emerging pressures or threats to the marine environment, identify gaps in
monitoring these pressures, and contribute towards a forward-looking strategy for monitoring
marine ecosystems.


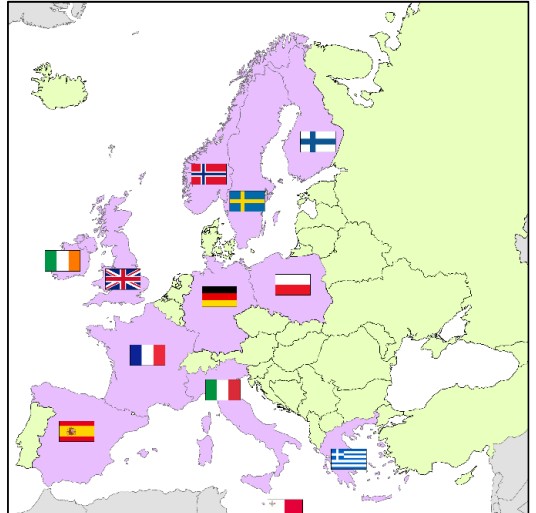


*Figure 1. The countries which participated in the poll were Finland, France, Germany, Greece,*
*Ireland, Italy, Malta, Norway, Poland, Spain, Sweden, United Kingdom.*

**2.    Methodology**
The opinion poll was designed as an online questionnaire, which could be completed over a
five-month period (29 June to 30 November 2016). The questionnaire was distributed to all
partners in the JERICO-NEXT project. Partners were tasked with being national
representatives and were asked to take responsibility for responding to the questionnaire
and/or to collect answers from colleagues, collaborators and responsible monitoring
authorities within their countries. The national representatives were also asked to forward the
questionnaire to the relevant authorities in countries which are not partners within JERICO-
NEXT.





Questionnaire development was informed by a review of existing literature on environmental
pressures and threats (e.g. EEA 2008a) and the outputs of the DEVOTES project (DEVOTES
2014). Threats to the marine environment were considered in terms of 'pressures' and
'impacts'. Pressures were described as the human activities which have impacts on
ecosystems or parts thereof (see Oesterwind et al 2016[11]), which is compatible with the
driver-pressure-state-impact-response (DPSIR) framework (Gabrielsen and Bosch 2003;
Elliott 2014).
**2.1.        Format of questionnaire**
The questionnaire (Figure 2, for more detail see supplementary material, S1) was developed
using Google Forms, and consisted of two linked forms. The first form was focussed on
obtaining the views of all respondents on the environmental threats in European waters and the
adequacy of current monitoring programmes. Maps were provided to ensure consistency in
participant selection of 'regions of interest' (see supplementary material, S2 and S3). For
questions related to pressures and impacts, respondents could select one or more responses
from lists provided. They could also add free text in order to provide detail or explanations of
their responses. Questions related to adequacy of existing monitoring programmes included
comments boxes for free text, to allow respondents to give their views on those monitoring
programmes which were not adequate or only partly adequate for addressing environmental
threats, and suggestions on how to improve the monitoring of the threats identified.
The second form was focussed on national monitoring programmes, with the aim of obtaining
a summary of sampling platforms used, variables measured, monitoring frequency and the
duration of the programme. This form included a section on data accessibility.
An invitation to participate in the poll and complete the questionnaire was sent to all partners
in JERICO-NEXT in June 2016 and subsequently forwarded to wider contact networks. It was
closed to responses in November 2016.

---

[11] *Pressures can be described as* 'a *result of a driver-initiated mechanism* (*human activity/natural process*) *causing an effect on any part of an ecosystem that may alter the environmental state'.* **Impacts** *can be defined as* '*consequences of environmental state change in terms of substantial environmental and/or socio-economic effects which can be both, positive or negative'.*


















Section 1 Environmental Threats and Monitoring
1. Participant Details:
    a. Name and contact details
    b. Institute/Affiliation
2. Region of interest (see Annex 2)
    a. Country
    b. Region
    c. Sub-Region
3. Review of threats per region
    a. Pressures: What are the main pressures from human activities that are affecting the environment in this area?
    b. Impacts: What are the impacts resulting from the pressures identified above?
4. Policy Purposes: What are the main policy or other drivers behind the monitoring programme/s in each region or sub-region? These may be international conventions, EU Directives, national policies, or other requirements.
5. MSFD Descriptors: The MSFD includes 11 qualitative descriptors. Please link the threats identified to these descriptors, or any others which may be relevant in the area.
6. Names of relevant monitoring programmes:
7. Adequacy of existing monitoring programmes: are they sufficient to assess the effects of the environmental threats in the considered area?
    a. How are they deficient?
    b. How could they be improved to better address the threats?

Section 2 Monitoring programmes
1. Country
2. Monitoring programme name
3. Is the program statutory/official or unofficial?
4. Variables measured
5. Platform types
6. Number of stations
7. Is monitoring regular or ad hoc?
8. Monitoring frequency
9. Start date
10. End date
11. End reason, if not ongoing
12. Monitoring stations (in separate spreadsheet).
13. Comments
14. Data access restrictions
15. Responsible organisation
16. Responsible person and details
17. Data source institute
18. Database to which the data are submitted
19. Are data flows to central databases up to date?
20. Web links to data

*Figure 2. Format of online questionnaire.*


**2.1.1. Data Analysis**
Once the poll was closed, responses were downloaded from Google Forms and stored in a MS
Access Database. Identifying information was removed from the responses to anonymise the
data. More than one response was received from some countries. Results on views or opinions
on environmental threats and impacts and on monitoring programmes were analysed by
country. Categorial responses were aggregated per country, counting each response if it
appeared at least once in the individual responses for the country.
Details of monitoring programmes and expert opinions on adequacy of monitoring
programmes were analysed for all respondents. Opinions were also analysed within each
country. Free-text responses from all respondents on the adequacy of monitoring programmes
were extracted to summarise all opinions given, and the suggestions for improving monitoring
programmes that were not adequate or partly adequate to address environmental threats.



To visualize the most common themes emerging from the questions on why monitoring
programmes were inadequate, word clouds were created using an online software tool (Wordle
2018), which emphasises the most common responses from individuals according to how many
times they are mentioned.

**3. Results**
**3.1. Respondents**
The online questionnaire was completed by representatives from 12 European countries
(Finland, France, Germany, Greece, Ireland, Italy, Malta, Norway, Poland, Spain, Sweden,
United Kingdom, Figure 1) representing different regional seas (Figure 3) and their sub-regions
(see supplementary material, S2 and S3). From some countries, responses were received from
more than one respondent, resulting in a total of 36 responses from the 12 countries. The most
responses (14) were received from the UK and covered territorial waters (12 nm) as well as
their Exclusive Economic Zone (EEZ) waters. Five responses were received from Greece, six
from France, two from Spain, and two from Malta. Many respondents were JERICO-NEXT
partners, but some were also from the wider European monitoring network.
**3.2. Views on environmental threats and impacts**
**3.2.1. Pressures from human activities**
Marine litter was identified as a pressure in 100% of the national responses (*Figure 4*). The next
most commonly identified pressures were shipping (92%), contaminants (92%) organic
enrichment (83%) and fishing (75%, *Figure 4*). These were followed by inorganic nutrient
enrichment and aquaculture (both 58%, Figure 4), dumping and extraction (50%), and
atmospheric inputs, dredging of biota and construction/obstruction (all 42%). Activities such
as mining, water abstraction, the oil and gas industry and coastal squeeze scored considerably
lower, at 10-23% of responses. Only one extra pressure was added to the list provided,
unexploded ordnance (UXO).
Respondents noted that the pressures affecting coastal and offshore areas were not the same.
Climate change related pressures (regime shift and ocean acidification) were considered to
have large potential for widespread harm and in all sea regions at least one respondent marked





regime change as an important pressure. Thermally-driven regime change was selected in a
greater proportion of responses than salinity-driven regime change.

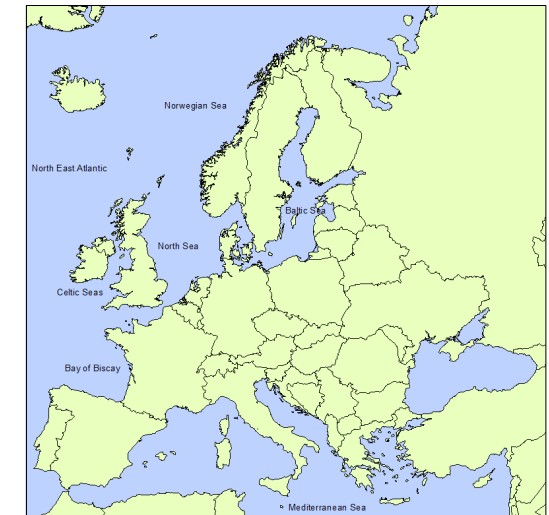

*Figure 3. The regional seas represented by respondents to the questionnaire (see supplementary*
*material for maps of regions [S2] and sub-regions of European seas [S3]).*

### 3.2.1. Impacts of the pressures identified

The majority of national responses (>70%) identified habitat loss or destruction, underwater
noise, and contamination as the main impacts of human activities on the marine environment
(Figure 5). Approximately 60% of national responses identified undesirable disturbance (e.g.
oxygen depletion), changes in sediment/substrate composition, changes in community
composition, harmful micro-organisms and invasive species as key impacts. Fifty percent
(50%) identified changes in primary production, changes in species range, population
change/depletion of standing stocks, biofouling, physical damage, changes in suspended
sediments/turbidity and mortality of marine life.





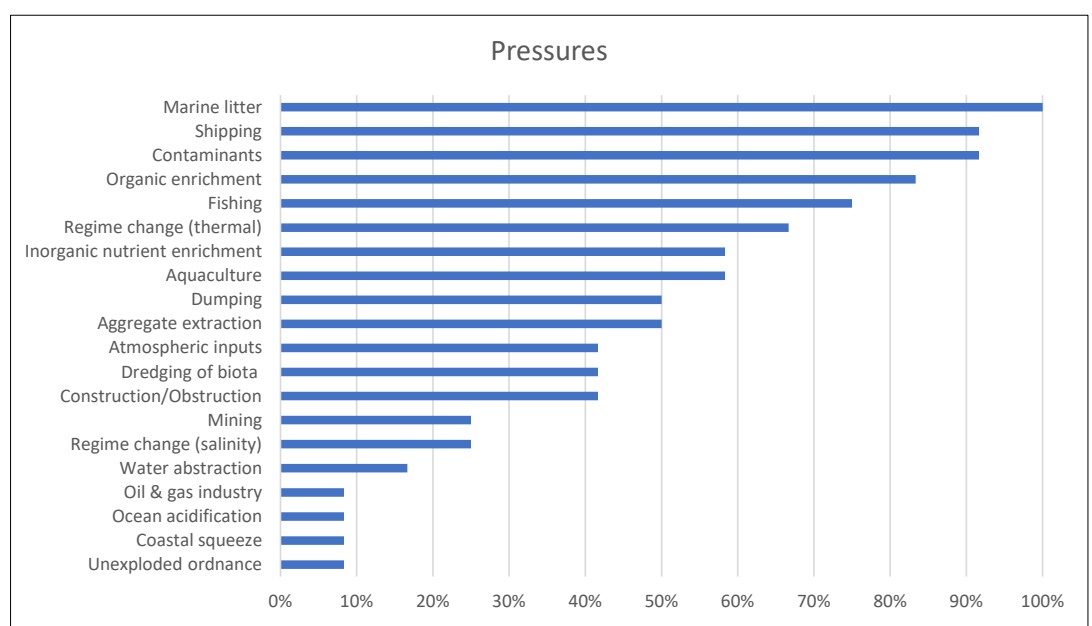

*Figure 4. Frequency of national responses on pressures affecting the marine environment.*

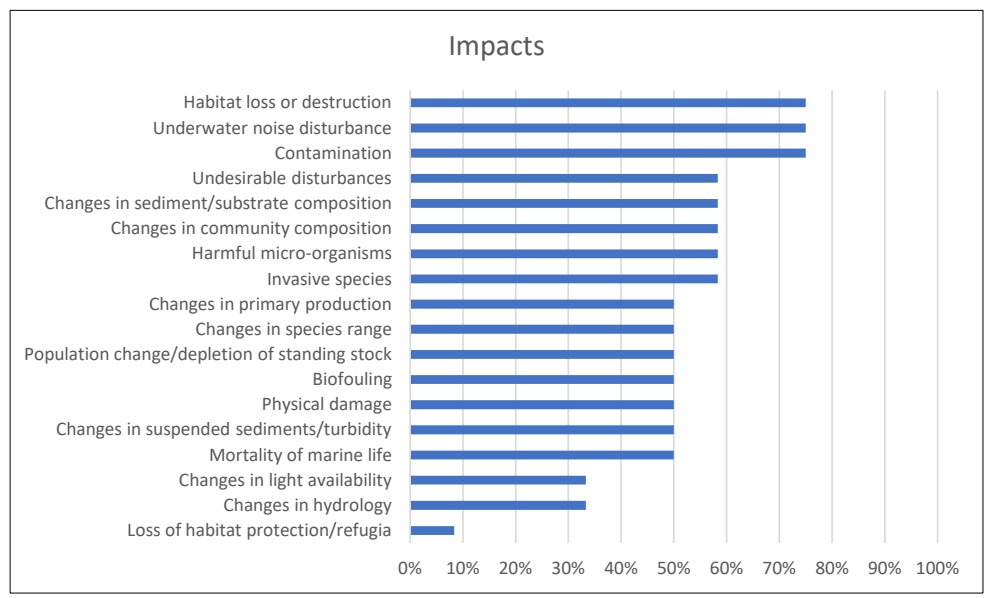

*Figure 5. Frequency of national responses on impacts affecting the marine environment.*





**3.3.    Views on the main drivers of marine monitoring**
**3.3.1. Policy purposes**
The majority of national responses (83%) identified the main drivers of monitoring of coastal
and offshore waters as the Water Framework Directive (WFD, EU 2000) and the Marine
Strategy Framework Directive (MSFD, EU 2008, Figure 6). Other EU directives were
identified but the proportion of national responses identifying these as policy purposes for
monitoring was relatively low. Twenty five percent (25%) of national responses included the
Urban Waste Water Treatment Directive and Nitrates Directive (Figure 6), and 17% included
the Bathing Waters Directive and the Nitrates Directive. Regional Seas Conventions were also
identified as drivers of marine monitoring, with OSPAR identified by 67% of national
responses and HELCOM identified by 17% of national responses. Local policy drivers were
identified by 58% of national responses, but no details were given.
Respondents were asked to link environmental threats in European waters to the descriptors
(D) in the MSFD (Figure 7; see EU 2008). Responses indicated that most threats (92%) affect
the biodiversity descriptor (D1, Figure 7). The next most frequent responses (83%, Figure 7)
were linked to descriptors for contaminants (D8), eutrophication (D5) and marine litter (D10).
Seventy five percent (75%) of threats could be linked to the energy descriptor (D11), 67% to
sea floor integrity (D6), hydrographic conditions (D7) and non-native species (D2), and 50%
to food webs (D4).

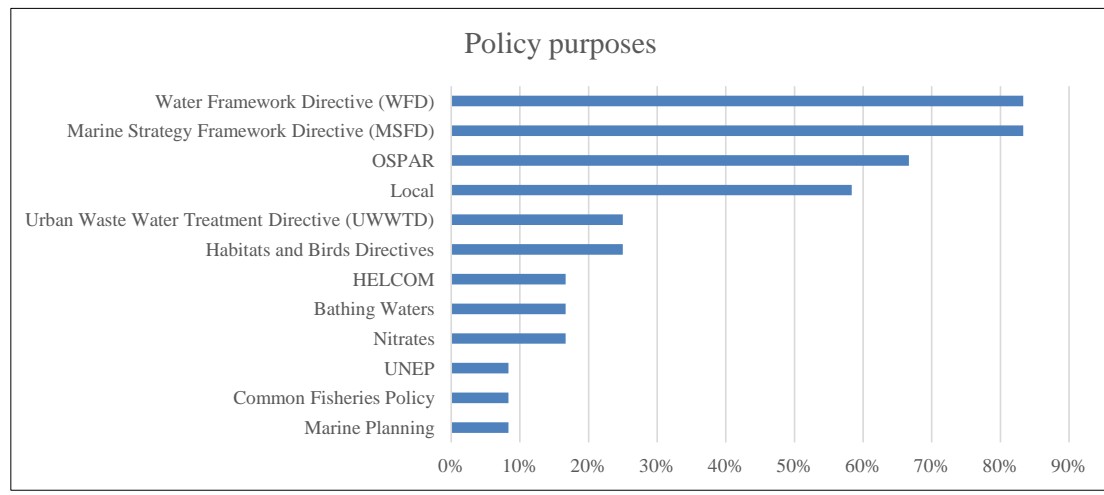

*Figure 6. Main policy or other drivers for marine monitoring.*

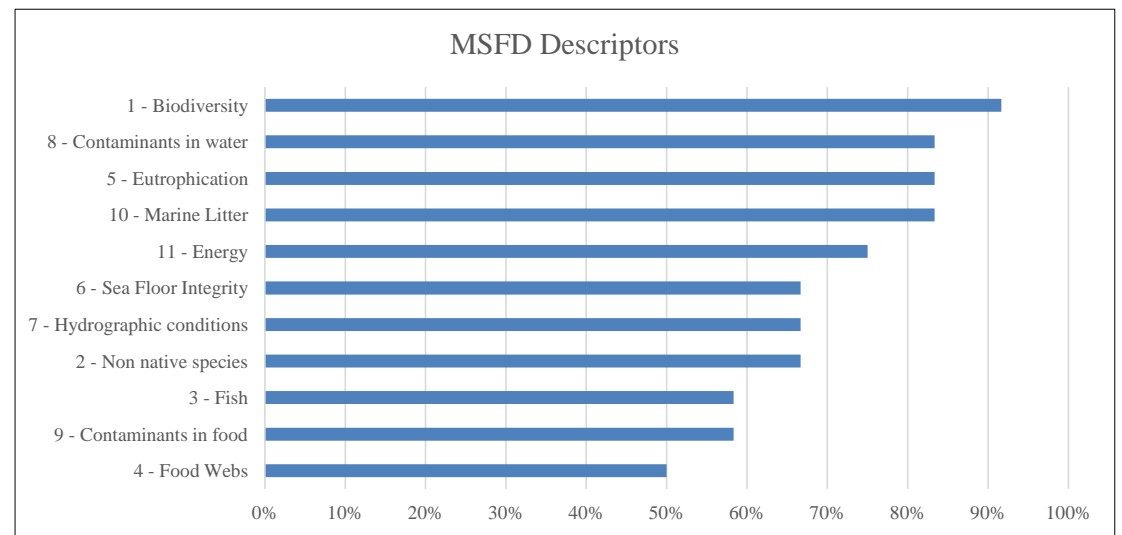

*Figure 7. MSFD Descriptors linked to environmental threats. The left axis shows the descriptor*
*number and name.*



### 3.3.2. Meeting requirements of policy drivers

Much of the monitoring towards older directives is now included in WFD monitoring
programmes implemented under River Basin Management Plans of Member States. These
results highlight that policy drivers may change over time but overall purposes may remain the
same or similar. Regional Seas conventions were also identified as key policy drivers of
monitoring programmes, with a greater proportion of responses for OSPAR than for
HELCOM.

### 3.4.   Monitoring Programmes in each country

In total, 36 responses on the monitoring section of the questionnaire were received from the 12
countries who participated in the online poll. Thirty-eight (38) monitoring programmes were
reported. More than half of these programmes were official or statutory programmes, and a
significant proportion (28%) were project based rather than statutory. These included the
Balearic Islands multi-platform observing system (SOCIB), UK BeachWatch litter project and
projects in Ireland (Smartbay observatory).





This is not a complete inventory of monitoring in Europe, but the responses provide examples
of a variety of monitoring programmes. Entries for the UK, Ireland and Greece appeared to be
relatively comprehensive.
**3.4.1. Monitoring: variables, platforms and frequency**
Most monitoring programmes were reported to measure temperature and salinity. A large
proportion of responses (39-45%, Figure 8) reported measurements of nutrients, chlorophyll
and dissolved gases, although not all parameters are measured at all stations in a monitoring
programme. Many variables, such as mammals, birds, biotoxins and marine litter are only
measured in specific monitoring programmes designed for the purpose. Some variables were
monitored in only a few monitoring programmes, e.g. sea level and contaminants, but this may
reflect the selection of responses received. Responses to the questionnaire indicated that marine
monitoring programmes provide less coverage of biological parameters (e.g. plankton 32%,
fish 18%, benthos 18%, macroalgae 11%, birds 3%) than physical water column parameters
(e.g. temperature, salinity, 58-61%) and chemical parameters (e.g. nutrients, dissolved gases,
45% and 39%).

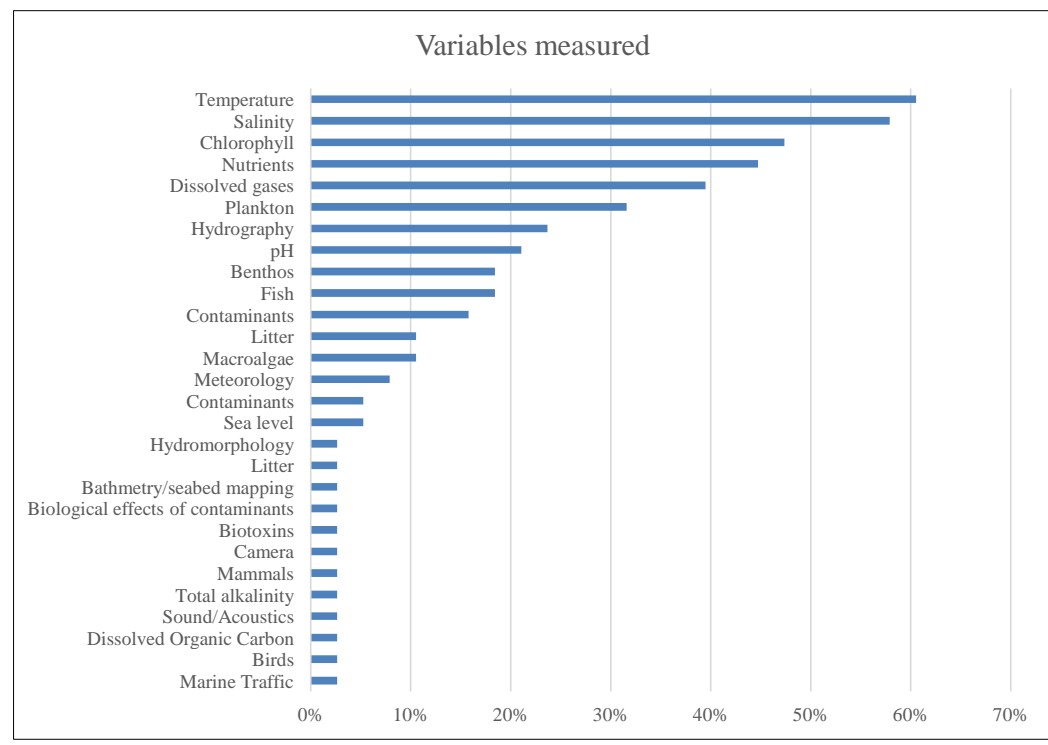

*Figure 8. Variables measured in marine monitoring programmes.*




Most monitoring programmes were reported to use a vessel as a monitoring platform (Figure
9), usually a research vessel or, for inshore monitoring, a small boat. Shore based monitoring
was also common (39%). The use of fixed platforms was indicated by 34% of respondents,
including those from Belgium, Greece, Ireland, Italy, Spain and the UK. The use of remote
sensing as a monitoring platform was reported by 21% of respondents (Figure 9). Remote
sensing is likely to complement other types of monitoring, rather than replace it, as in situ data
is needed for validation and it is limited to surface monitoring of particular parameters. Other
innovative and emerging technologies, such as autonomous vehicles, FerryBoxes and 'other'
(e.g. profiling floats) were included in ≤11% of the responses (Figure 9).

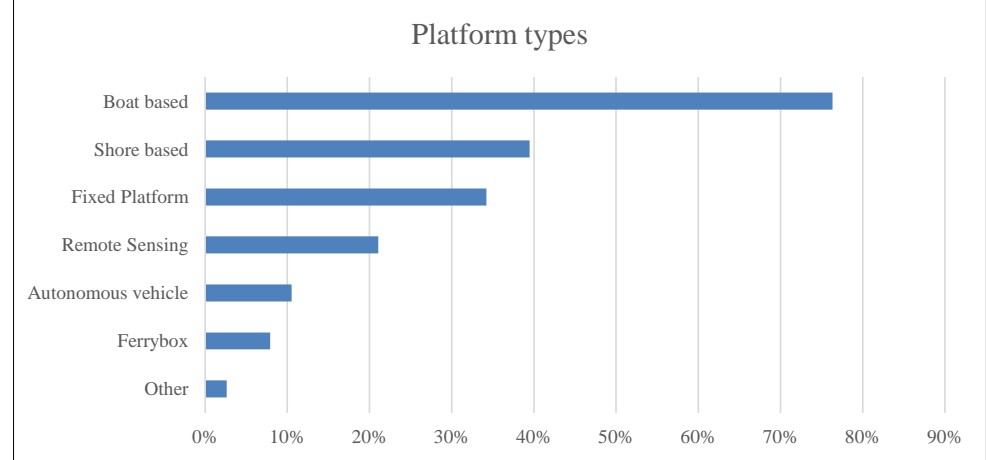


*Figure 9. Platform types used in marine monitoring.*

Responses to the questionnaire indicated that monitoring frequency (*Figure 10*) is variable. The
highest proportion of responses (34%) was for continuous monitoring (e.g. from fixed
platforms, moorings or gliders). Several monitoring programmes were reported to have only
annual monitoring, but to be comprehensive in terms of parameters and spatial coverage.
Monitoring programmes incorporating fixed platforms or gliders were more restricted in terms
of spatial coverage.

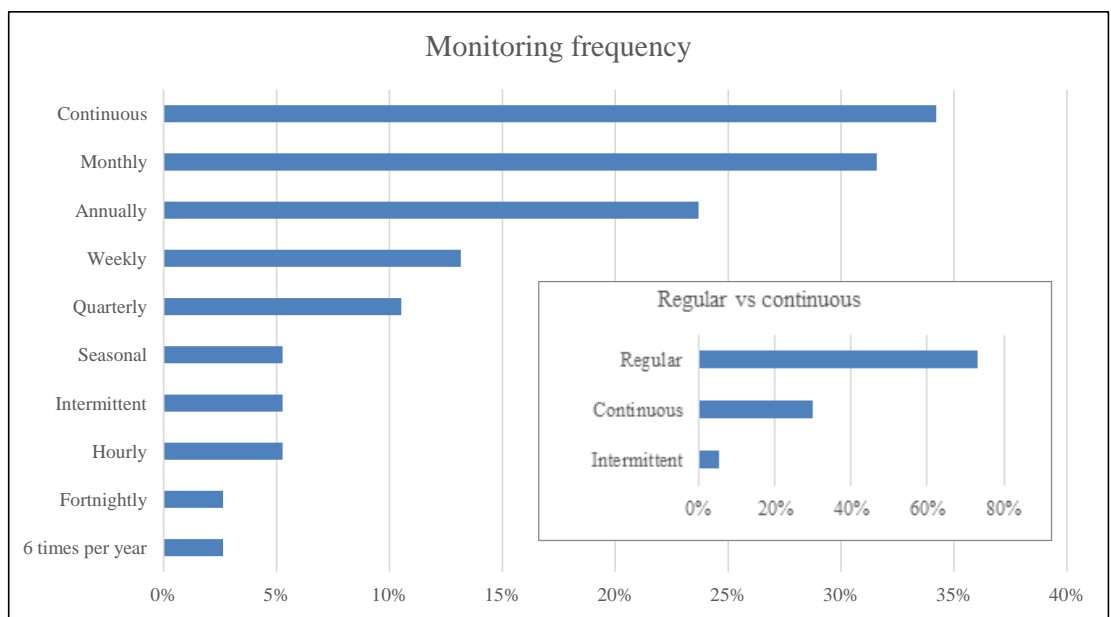

*Figure 10. Frequency of monitoring. The main graph shows results for all options given in the questionnaire. The inset combines these into three categories: continuous and intermittent are the same as in the main graph, regular = all other options combined.*

### 3.4.1. Sustainability of monitoring programmes

Responses to the questionnaire showed that 68% of the monitoring programmes have been running for longer than 10 years. The longest programme reported was the continuous plankton recorder survey, by the Sir Alister Hardy Foundation for Ocean Science (SAHFOS), which has been running since 1931. Several French and Scottish monitoring programmes were reported to have been running for approximately 30 years. One respondent included a monitoring programme which ended due to lack of funding; it is likely there were many more such cases which were not reported.

### 3.4.2. Data access

The majority of respondents (71%) reported that their monitoring programmes had no restrictions on data access. Where data access is restricted, most programmes make the data available on request, subject to information on the intended purpose or use of the data, signing of a licence agreement, and/or requirements to acknowledge the source of the data (e.g. through the use of data DOIs [digital object identifiers]).





Respondents reported that data were submitted most commonly to local/national databases, but
frequently also to ICES databases, EMODnet or Copernicus. For the majority of programmes,
data flows to these central databases were considered to be not up-to-date, indicating that not
all monitoring data are available centrally, or that there is a time lag in submission of data.
**3.5.        Gaps identified in current monitoring programmes**
In terms of providing the information required to monitor environmental threats, 12% of all the
respondents to the questionnaire considered monitoring programmes to be adequate, while 28%
indicated that monitoring programmes were not adequate and 60% considered monitoring
programmes to be partially adequate (Figure 11).

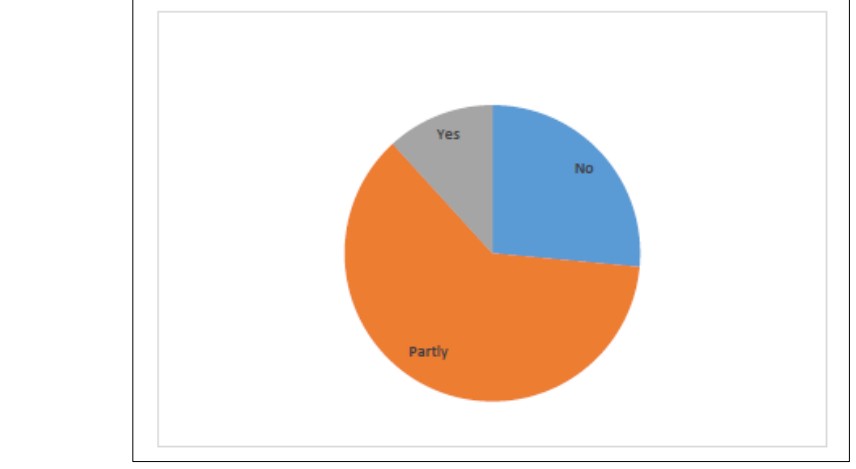

*Figure 11. Proportion of all respondents who considered their monitoring programmes to be*
*adequate (Yes), inadequate (No) or partly adequate (Partly).*
Where there was more than one respondent per country, responses were varied (Figure 12),
with the majority of responses indicating inadequate monitoring. In the UK, for example, where
14 responses were received, most responses (57%) were that monitoring was partly adequate,
and 29% were that monitoring was not adequate. Two respondents (15%) felt that monitoring
programmes were adequate. In France, where six responses were received, the majority (83%)
considered monitoring was not adequate, and the remaining 17% felt it was adequate. In
Greece, four out of five respondents (80%) felt monitoring was not adequate, and one
considered it to be partly adequate. In countries with two responses (Italy, Malta and Spain),
one indicated that monitoring was not adequate while one felt it was partly adequate. In





countries with one respondent, responses were mostly that monitoring was partly adequate
(Finland, Ireland, Norway, Sweden). In Poland, the national representative reported that
monitoring was adequate.

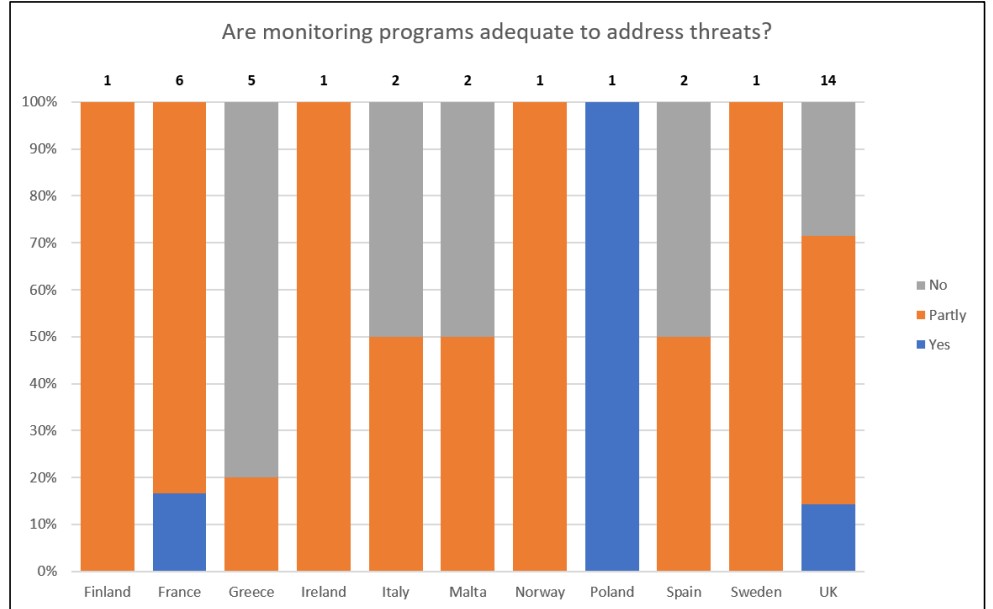


*Figure 12. Responses by country showing the proportion of respondents who considered their*
*monitoring programmes to be adequate (Yes), inadequate (No) or partly adequate (Partly). The*
*number of respondents per country ranged from 1 to 14 (see numbers in bold).*

### 371  3.5.1. Where monitoring is not adequate

Responses were focussed around a few key issues (see Figure 133) which appeared to be related
mostly to insufficient resolution in time and space, insufficient data or parameters measured,
and lack of integration (e.g. of monitoring programmes, indicators and descriptors).
A number of respondents stated that there is insufficient monitoring for some of the MSFD
descriptors. These descriptors included biodiversity, food-webs, marine litter (including micro-
plastics), underwater noise, emerging contaminants, and emerging pollutants. However, no
details were given. It was noted that coupling between physics and biology in response to
environmental pressures is typically not included in monitoring programmes focussed on
individual descriptors. One respondent indicated that methodologies and approaches were not



state-of-the-art, for example, the focus during benthic sampling was on taxonomy instead of
ecosystem functions and services.



Figure 13. Key words used in views on partially adequate or inadequate
monitoring programmes. Font sizes indicate the most common responses from
individuals according to how many times they are mentioned.

Two respondents highlighted concerns about the links to policy drivers, suggesting that
monitoring was reactive rather than proactive. One of these respondents commented that
monitoring programmes develop to respond to pressures and impacts. The other highlighted
concerns related to unexploded ordnance, for which there seems to be very little political or
commercial interest in finding and making safe dumped munitions, until a person or marine
life is found with injuries or abnormal growth.
Examples of monitoring programmes with low spatial resolution were given for sub-regions of
Mediterranean Sea; point source monitoring of contaminant inputs, controls and
improvements; benthic habitats for the wider environment, and deep-sea areas. Examples of
inadequate monitoring of parameters were given for the Mediterranean Sea: zooplankton,
phytoplankton compositions, marine mammals, reptiles, birds, invasive species, marine litter,
and contaminants in sediment and biota.



## 4. Improving monitoring programmes

The respondents gave a number of suggestions for improving monitoring programmes considered to be not adequate or partly adequate. These were focussed on improved design of monitoring programmes and increased effort, observations and research, as follows:

1. To develop monitoring programmes that are fit-for-purpose and meet policy needs, through working with policy end-users. For example:
   - to meet requirements for spatial representativeness of data.
   - for assessing benthic habitats in the wider environment (beyond Marine Protected Areas).
   - work planned in the Welsh sector of the Irish Sea to develop offshore renewables industries will result in considerable local impact. Appropriate monitoring needs to be installed to measure a range of physical and ecological variables in order to assess impact.
2. To take into account regional or national specificities. For example: sub-regions of regional seas.
3. To assess availability of information in relation to pressures. For example:
   - In Wales, there is a recognition that there is incomplete information on the fishing pressure on inshore fisheries. Steps are being taken to introduce inshore Vessel Monitoring Systems that will automate the process of gathering better information on fishing activity and seabed disturbance.
4. To make better use of low-cost biochemical sensors on low-cost platforms.
5. To increase observations in time and in space, and include parameters that provide information on ecosystem function.
6. To monitor marine waters extending beyond the coastal zone and adding more biological and chemical parameters. For example: zooplankton, microbes, marine mammals (Mediterranean Sea).
7. To develop coordinated and integrated monitoring programmes.
8. To increase effort to improve monitoring of:
   - biodiversity components not yet monitored
   - poorly covered habitats
   - small plankton
   - monitoring of beaches in some countries
9. To systematically monitor marine litter and noise.
10. To implement systematic monitoring based on rigid baseline ecological assessment (at small local scales, e.g. Mediterranean Sea)
11. To increase monitoring in high-risk areas.
12. To have consistent and routine fixed-point monitoring (e.g. Malta island).
13. To develop a limited number of long-term monitoring sites in remote areas to monitor changes in baseline conditions (chemistry, ecotoxicology, and ecosystem structure) in response to climate change/acidification, and diffuse inputs.



14. To incorporate newer threats (e.g. phosphorous-based flame-retardants, microplastics, noise) into regular monitoring.

15. To be more proactive regarding threats likely to cause harm to or changes in biota, e.g. unexploded ordnance (UXO).

16. Deployment of additional observatories for the assessment of biodiversity and water quality. JERICO-RI may contribute to filling in the gap, especially for water quality and biodiversity of phytoplankton.

17. To include flexible research/investigative monitoring to increase knowledge of specific impacts.

18. To secure funding for long-term monitoring programmes.

## 5. Discussion

### 5.1. Polling Approach

The opinion poll carried out during this study had a limited number of participants, as it was targeted towards scientists and managers with the relevant expertise and experience in European countries adjoining regional and/or sub-regional seas. In order to minimise bias which might be introduced by some countries providing more individual responses than other countries, project partners were expected to develop national responses, and were given approximately six months to do so. Where there was more than one response from a country, results on views or opinions were combined to represent a national view; responses on monitoring programmes were not combined, as these were considered to provide useful detail on gaps in monitoring, and no monitoring programmes had duplicate responses.

Despite a number of limitations in the polling approach, responses provided valuable insights on the environmental pressures and their impacts, and on gaps in monitoring the impacts.

### 5.2. Drivers of marine monitoring

Most national responses were focussed on policy drivers such as EU Directives and regional conventions based on the ecosystem approach. These responses are likely to have been influenced by the overall context of the JERICO-NEXT project and its emphasis on biogeochemical processes and the coupling between physics and biology. Responses may also have been influenced by the drop-down list of options from which to select answers, although the option was given to add responses.





Interestingly, local drivers scored quite highly. No details or examples were given by any of
the respondents but may include monitoring towards impact assessments for a variety of
reasons, such as development of local fisheries or recreational activities, or to meet
conservation objectives (e.g. for marine protected areas). Such monitoring would be included
under policy drivers such as the Habitats and Birds Directives or Marine Planning, and
relatively few responses (≤25%) indicated these as drivers for marine monitoring.  It is possible
that local drivers included research projects or programmes, but this seems unlikely as the poll
was focussed on monitoring rather than research. This highlights a potential weakness of the
aims of this study and indeed the JERICO-NEXT project, as it did not include an objective to
identify gaps in understanding, and how to provide better linkages between research and
monitoring. Certainly, ongoing national monitoring programmes are focused on reporting to
directives and international obligations, and not to contribute to better understanding of the
possible impacts of the threats.
Complex linkages between pressures and impacts and the cumulative effects of multiple
pressures are not currently well addressed by any of the reported monitoring programmes. The
MSFD was intended as a holistic approach to assessments, but descriptors are currently
assessed separately. Developments are underway to move assessments towards a more
integrated cross-disciplinary ecosystem approach (e.g. OSPAR[12]; EEA 2011; EEA 2015b;
HELCOM 2018). This will require more co-ordinated monitoring across descriptors, and a
focus on acquiring long-term data sets, particularly for addressing cross-cutting issues such as
climate change and ocean acidification (e.g. Tett et al 2013). Responses indicating that a
number of monitoring programmes have been running for more than 10 years are extremely
positive, providing data to allow the detection of temporal trends on pressures and their impacts
on the marine environment. Evidence that a significant proportion of monitoring is largely
project-based rather than statutory, indicates some degree of risk to the sustainability of
monitoring. EuroGOOS conducted a survey of sea level monitoring and found similar issues;
less than half of the organisations responding considered that there were no funding issues for
tide gauges and many had reduced funding or uncertain future funding (EuroGOOS 2017).
With the majority of responses to the online poll indicating that the main policy drivers of
current monitoring are the MSFD and WFD, rather than earlier directives such as the UWWTD

---

[12] See https://oap.ospar.org/en/ospar-assessments/intermediate-assessment-2017/introduction/assessment-
process-and-methods/; and https://oap.ospar.org/en/ospar-assessments/intermediate-assessment-2017/chapter-6-
ecosystem-assessment-outlook-developing-approach-cumul/

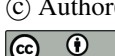



and the Nitrates Directive, it is clear that policy drivers and requirements for meeting policy
needs change over time. The findings also highlight that monitoring programmes should be
underpinned by high-level scientific objectives, and that research and monitoring should be
well integrated. Data sharing, such as through the JERICO-NEXT research infrastructure and
coastal observatories, is vital to current and future integration of research and monitoring
(Farcy et al 2019). Furthermore, the availability of data at local and regional scales is essential
for development of future monitoring and assessment approaches, particularly as new
technologies and assessment tools are developed and become more readily available (e.g. Borja
et al 2019; García-García et al 2019).
**5.3.        Views on environmental threats and impacts**
Respondents were provided with comprehensive lists of key environmental threats and impacts
informed by previous studies, with an option to add to the list. One item, UXO, was added to
the list of pressures by one country. This pressure was considered to be outside the scope of
the JERICO-NEXT project but may be useful in other contexts. No new items were added to
the list of impacts in the national responses.
Key pressures or threats to the marine environment due to manageable human activities (i.e.
>70% of national responses) were considered to be marine litter, shipping, contaminants,
organic enrichment, and fishing.
Key impacts of the threats to the marine environment (i.e. >70% of national responses) were
identified to be habitat loss or destruction, underwater noise, and contamination. Sixty percent
(60%) of national responses identified key impacts to be undesirable disturbance (e.g. oxygen
depletion), changes in sediment/substrate composition, changes in community composition,
harmful micro-organisms and invasive species.
**5.4.        Monitoring programmes**
Most respondents were of the view that current monitoring is partially adequate or not
adequate. The range of views given between and within countries suggest that a broad spectrum
of participants responded to the questionnaire. These views likely reflect different experiences
of respondents in their areas of expertise and in their countries.
Key issues identified (i.e. insufficient resolution in time and space, insufficient data or
parameters measured, and lack of integration) indicate the gaps in monitoring. Suggestions for



improved monitoring programmes were targeted at these gaps, and need to be considered in
detail to feed into the science and monitoring strategies. These issues are discussed below.
**5.5.        Resolution in time and space**
The scale of impacts varies widely, with some activities, such as construction of a wind farm
having a potentially high impact on a small area, whereas activities such as fishing are more
widespread. The impact of human activities also depends on the vulnerability of the habitat in
question. For example, in the southern Celtic Sea, fragile benthic habitats with cold-water
corals are highly impacted by sea floor activities. Some impacts, such as noise disturbance,
depend on the intensity of the activity, and will be concentrated in areas with high shipping
activity, or during periods of construction.
Countries such as the UK adopt a risk-based monitoring approach, which was considered to
result in fragmented monitoring. Examples of low spatial resolution were given for the CPR
survey, one of the key plankton datasets, where spatial gaps exist throughout EU waters. Spatial
resolution was also considered to be low for some habitats, as not all habitats are covered by
monitoring programmes, and for monitoring of marine litter and non-native species.
In terms of spatial resolution, other responses indicated that not all parameters are monitored
adequately. Responses included biogeochemical parameters, although no examples were given,
and zooplankton. The WFD does not require zooplankton monitoring, but some indicators
under the MSFD do require information on zooplankton. Although phytoplankton is monitored
inshore, the data are disparate and mainly used to report on potential health issues due to toxin
producing algae.
For temporal resolution, examples were given for a number of threats where the monitoring
period was considered to be inadequate. For example, for statutory monitoring of impacts
such as those from dredging and disposal, monitoring is often over time scales which are too
short (2-5 years) to properly assess the impacts on the biological communities. This also
applies to seabird and cetacean monitoring, which is out of the scope of JERICO-NEXT.
Some monitoring programmes may be inadequate in terms of temporal frequency: 24% of
monitoring programmes reported had annual monitoring, which may fail to detect impacts
throughout the year. Monitoring frequency is likely to be strongly influenced by platform
types, with increasing use of fixed platforms, moorings or gliders giving a high proportion
(34%, Figure 10) of responses for continuous monitoring. Certainly, platforms such as
moorings can provide high-frequency temporal resolution (e.g. Mills et al 2005; Greenwood





et al 2010) for the parameters they measure, predominantly physical and chemical parameters
(such as temperature, salinity, light, dissolved oxygen) with biological parameters limited to
phytoplankton fluorescence or chlorophyll.
Addressing the issue of scales is essential in establishing a future pan-European monitoring
program, particularly for biological parameters. Monitoring these parameters is more limited
than for physical parameters. Reasons for this include that:
(1) The types of biological data that can be automatically or semi-automatically acquired is

low despite recent technological developments (including those achieved within FP7-

JERICO and JERICO-NEXT), which limits the spatio-temporal coverage of

biological/biogeochemical data sets

(2) Miniaturization of sensors allowing for implementation on platforms such as AUVs and

floaters is more feasible for physical and chemical parameters, which results in better

spatial and synoptic coverage

(3) Scaling-up from "point" observations to wider areas most often relies on modelling.

Physical models are more advanced than biogeochemical and biological models, which

also increases the importance of scales of biological observations.

### 5.5.1. Small scale threats/disturbances

The majority of threats impact at relatively small spatial and temporal scales, at least initially.
Examples include the accumulation of marine litter, the development of harmful algal blooms,
and the invasion by non-native species, which occur locally in the first instance, as influenced
by point sources and the characteristics of the abiotic and biotic components of the
environment. In these examples, there is no initial discrepancy in spatial scales between
monitoring and threats/disturbances. However, the number of monitored habitats clearly
remains too low, as indicated by responses to the questionnaire.
Monitoring effort should be sufficient in time and space to: (1) detect the effects of new
threats/disturbances acting in different locations within the same habitat, (2) assess the
consequences of an identified threat/disturbance at larger scales, and (3) assess cumulative
effects of multiple threats.

### 5.5.2. Large-scale threats/disturbances

Some environmental threats act over large spatial scales, such as thermal regime change or
ocean acidification. There is a discrepancy between the (large) spatial extent of the



threat/disturbance and the (small) scale at which the monitoring is performed (station). This
may be addressed to some extent by (1) the use of mobile monitoring techniques such as
FerryBoxes which allow for large geographical coverage, albeit on a limited time-scale, and
(2) the fact that only a small number of fixed monitored sites is required to monitor this kind
of threat disturbance. Factors to consider include that:
(i) Different biological communities may not be affected in the same way by the same level
of a given (widespread) environmental pressure. Grémare et al (1998) and Labrune et al
(2007), for example, clearly showed that in the Gulf of Lion, the composition of the two
shallowest communities (i.e. littoral fine sands and littoral sandy muds) are most affected
by climatic oscillations. A sound assessment of large-scale threats/disturbances at the
reporting scales should therefore not rely on the sampling of a single, or even a limited
number of habitats. Conversely, the monitoring strategy of large-scale
threats/disturbances should ideally encompass all the habitats present in the reporting
area.
(ii) The representativeness of monitoring data is often limited. For example, highly mobile
fauna (e.g. marine mammals or birds) are often used as proxies for large scale
threats/disturbances because they can be found over large spatial scales and because, as
for predators, their ecophysiology and/or population dynamics tolerate a large set of
ecological processes. The probability of these organisms being sampled with confidence
is directly proportional to the sampling effort and to their relative accessibility. Current
monitoring resources currently deployed in the UK, for example, do not have the power
to detect trends in all seabird and cetacean species or identify the drivers of their
population change. A similar example was given for Malta, where only the most
accessible marine bird nests are currently monitored as part of the seabird monitoring
program.
**5.5.3. The real world: a mixture of threats/disturbances at small and larger scales**
At the scale of global coastal marine ecosystems, several environmental pressures act
simultaneously, each having its own spatial resolution and temporal dynamics. Halpern et al
(2008) and Crain et al (2009) found that no fewer than five pressures overlap anywhere in the
world's oceans. Potential cumulative and/or interactive effects need to be addressed, for
example by considering that:



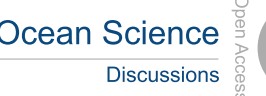

(i) Monitoring should be based on the largest spatial entity within which the comparisons of community compositions are sound, e.g. habitats or ecohydrodynamic regions (van Leeuwen et al 2015).

(ii) The monitoring of each habitat or region should include a sample size large enough to allow for the detection and the variability in the effects of small- and large-scale threats/disturbances.

(iii) Within a given reporting area, a monitoring program should include the highest possible number of relevant habitats in order to facilitate the detection of new small-scale threat/disturbance and the upscaling of large-scale threat/disturbance effects.

Such monitoring programmes would require considerable effort, highlighting the need to define/characterize relevant environmental.

The feasibility of the different suggestions for improved monitoring needs to be considered. This includes the identification of 'new technologies' and how best to incorporate them into monitoring programmes. Projects such as JERICO-NEXT work to harmonise new technologies which may be able to solve some problems of scale through high-frequency monitoring. For example, instruments such as flow cytometers and multispectral fluorometers can measure continuously on research vessels or buoys and so provide good spatial and temporal coverage. However, integrating these data types into existing monitoring presents several challenges: data may be in a very different format (continuous versus discrete samples, functional groups vs taxa), adopting new methods may affect the integrity of long time series, or there may be difficulty gaining acceptance and confidence in new methods. Similar challenges exist with using remotely sensed data instead of field measurements (e.g. for turbidity, chlorophyll), and these also still requires ongoing in situ measurements for validation (De Cauwer et al 2004).

## 6. Conclusions

This study consolidates the main conclusions from the Dobris Assessment (EEA 1995) and more recent studies (e.g. EEA 2008a, b; EEA 2015a; DEVOTES; Tett et al 2013; Zampoukos et al 2013; Garcia-Garcia et al 2019), highlighting the need for improved monitoring of environmental threats in European coastal environment.



Responses to the JERICO-NEXT questionnaire highlighted key gaps between the
environmental pressures or threats and their impacts, and the monitoring of these impacts. The
key findings were that:
• Ongoing national monitoring programmes are focused on reporting to directives and
international obligations, and not to contribute towards better understanding of the possible
impacts of the threats.
• Monitoring programmes are largely inadequate in terms of spatial or temporal resolution,
and for the assessment of emerging threats.
• Monitoring of biological parameters is generally inadequate, with insufficient focus on
coupling between biological and physical or chemical parameters.
• New technologies such as remote sensing, FerryBoxes, and gliders could help fill some
spatial and temporal gaps in monitoring.
• Submission of monitoring data to central databases needs to be improved to ensure that
monitoring data is available centrally.
• Issues of scale need to be addressed in fit-for-purpose monitoring programmes.
• More integrated cross-disciplinary approaches will require more co-ordinated monitoring
across descriptors.
• Although some monitoring programmes address multiple pressures, there is scope for more
harmonisation through improved monitoring design to create programmes which are fit for
multiple purposes.

The JERICO-RI has high potential to fill in some of the observation gaps, especially related to
physical and biogeochemical parameters, and the coupling between biology and physics across
scales needed for integrative understanding. Through the JERICO-NEXT project, the JERICO-
RI could become a major contributor towards future coastal monitoring programmes, through
the elaboration of a science strategy which would pave the way to a better integration of
physical, chemical and biological observations into an ecological process perspective. The
particular challenge of simultaneously observing physical, chemical and biological parameters
for assessments of complex coastal processes remains an open issue in relation to the temporal
scale of sampling. This will be addressed in the JERICO-NEXT science strategy under
development.



Certainly, one of the main challenges for the European marine research community is to
increase the consistency and the sustainability of dispersed networks and infrastructures by
integrating them within a shared pan-European framework. The long history of national
monitoring programmes which have been expanded, modified and developed over time,
together with methodological differences between nations, results in difficulties for the
integration and holistic assessment of the data (at a regional sea level) which the JERICO-RI
may contribute towards solving.

**7.    Acknowledgements**
This project has received funding from the European Union's Horizon 2020 research and
innovation programme under grant agreement No. 654410.

**8.    Author contributions**
SP, KC, DD, AG and GB designed the questionnaire. KC downloaded and analysed the results.
SP and KC prepared the manuscript with contributions from all co-authors.

**9.    Competing interests**
Six of the authors declare that they have no conflict of interest. VC is a member of the editorial
board on other topic areas in the Special Issue.

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
