# Peer review of "Marine monitoring in Europe: is it adequate to address environmental threats and pressures?"

_Ocean Science, 2019_

## Referee Comment (RC1) · Sander Wijnhoven (Referee) · 5 Sep 2019

Review 'Painting et al' OS-2019-75: By Sander Wijnhoven (Ecoauthor; www.ecoauthor.net) 5 Sept 2019

Marine monitoring in Europe: is it adequate to address environmental threats and pressures?

Suzanne J. Painting, Kate A Collingridge, Dominique Durand, Antoine Grémare, Veronique Créach, Christos Arvanitidis, Guillaume Bernard

The paper of Painting et al. (Marine monitoring in Europe: is it adequate to address

environmental threats and pressures?) gives a nice overview of the results of an inventory to identify whether current marine monitoring is sufficient to address environmental threats and pressures. This is an important question that should in the first place be asked by national governments and in the second place by those driving policy concerning marine environmental health by the EU (responsible Directives) and Regional Seas Conventions. Although the answer is not really a surprise (current monitoring of threats is partially adequate or not adequate); it is good to have it verified, to see where the major points of concern are, and to use this paper as one of the starting points for improvement. Although there are a few aspects (basically terminology and points of discussion as indicated below) that could/should be clarified, the methodology seems to be solid and transparent. The paper is generally well written and well structured.

P1-L18-19 (Abstract): 'Regime shift was identified as a pressure . . .'. - Is regime shift a pressure or the effect of a pressure, which might have an impact on itself?

Ok, I learn from the questionnaire that changes in temperature or salinity conditions are meant and not necessarily 'regime shifts' as "large, abrupt, persistent changes in the structure and function of a system". Thermal pollution or salt or freshwater discharge is definitely a pressure. Changes in thermal or salinity condition can also be effect of extractions, obstructions, global change, etc., and there with an impact. Clarify 'regime shift' as used here and elsewhere.

(The file with the supplementary material referred to is (still) named 'What is your gender'. – Please change the file-name.

P1-L19-L23: What is the difference between the main impacts and the key impacts; is it possible that those key impacts are actually effects of impacts? Clarify 'key impacts' as used here and elsewhere.

P8-Ch3.1: Talking about 36 responses you mean '36 individuals who filled in form S1', that probably came with in total a huge amount of forms S2 and S3 I guess? How do you prevent getting a skewed/biased view on the theme as for instance almost 40% of

the responses is from the UK? It seems that it is via working with 'national responses' as I gradually start to understand. Is there a pattern in the number of pressures or impacts identified per country with the number of responses per country than?

P8-L198: '100% of the national responses' – What does this mean? Are there responses considered not national (e.g. from researchers/people not working for the government)? Or does this mean that when one of the responses from a country includes the pressure (independent of the number of responses), it is considered to be identified for that country?

P13-Ch3.4.1: 'Responses to the questionnaire indicated that marine monitoring programmes provide less coverage of biological parameters than physical water column parameters and chemical parameters.' – Is this indeed the result per programme? Than the question arises about coverage (does monitoring take place at the same scale or with similar numbers of stations, or are there singular programmes covering large areas compared to several few station programmes for other aspects)? Is the presence or absence of monitoring of certain parameters at a national-subregional sea (or finer scale) level not a better indicator? Or are we only talking platform-based monitoring? Please discuss this?

P16-L358: Talking about percentages based on only 6 cases is a bit strange (17% is one respondent); at least do both.

P17-L372: 'Figure 133' should be 'Figure 13'. P26-L642-643: 'Such monitoring programmes would require considerable effort, highlighting the need to define/characterize relevant environmental'. - What do you mean?

---

## Referee Comment (RC2) · Anonymous Referee #2 · 4 Nov 2019

This paper presents the results of an opinion poll on marine monitoring needs and gaps based on a questionnaire filled by a number of scientists (36) from 12 European countries involved in the EU-project JERICO- Next. The topic of the questionnaire is interesting and timely. The paper is well written and perfectly fine as a project deliverable, and as a way to advertise the project. To make it more interesting to a broad and specialised scientific community, a better balance should be taken among the topics stated in the abstract by giving more space to an updated synthesis of the literature and to an exhaustive and informative overview of the monitoring operations in place. This approach would possibly allow to improve and sharpen the conclusions which are presently too general. As acknowledged by the Authors, the topic of the questionnaire

has been addressed several times in recent years, but the most recent and relevant contributions to OceanObs '19 (published in Frontiers in Marine Science) are not considered. The respondents had been given the task to act as national representatives, but it is not clear how this worked, because multiple questionnaires were filled for several countries, with 14 only from UK. This results in a lack of balance among countries and the bias that only countries and scientists participating in the project were directly involved, while it is hard to assess the actual coverage of those opinions not even in individual countries. The other weakness is that the large part of the paper is based on opinions and partial overviews of the various issues. Although coming from qualified people and potentially interesting, it is hard to check their actual soundness and the completeness of the information. Among the improvements suggested in section 4, I was surprised not to find any references to advanced molecular technologies. The most interesting part could have been the overview of the Monitoring Programmes in each country, but admittedly the reported programmes were a subsample rather than an exhaustive inventory.

---

## Author Comment (AC1) · 2 Dec 2019

Author response: We would like to thank the reviewers for their comments and the opportunity to address these in the manuscript. Our responses to comments are shown below (in red text in our document). We have tracked changes in the MSWord version of the revised manuscript. These changes also address the comments of the other reviewer/s. We have revised our Conclusions (Section 6): we removed repetition and moved some of the text to Section 4, which was then edited to improve the synthesis of responses. Changes in Sections 4 and 6 were quite considerable, and tracking was removed.

[Figure]

Reviewer 1: The paper of Painting et al. (Marine monitoring in Europe: is it adequate to address environmental threats and pressures?) gives a nice overview of the results of an inventory to identify whether current marine monitoring is sufficient to address environmental threats and pressures. This is an important question that should in the first place be asked by national governments and in the second place by those driving policy concerning marine environmental health by the EU (responsible Directives) and Regional Seas Conventions. Although the answer is not really a surprise (current monitoring of threats is partially adequate or not adequate); it is good to have it verified, to see where the major points of concern are, and to use this paper as one of the starting points for improvement. Although there are a few aspects (basically terminology and points of discussion as indicated below) that could/should be clarified, the methodology seems to be solid and transparent. The paper is generally well written and well structured. P1-L18-19 (Abstract): 'Regime shift was identified as a pressure . . .'. - Is regime shift a pressure or the effect of a pressure, which might have an impact on itself? Ok, I learn from the questionnaire that changes in temperature or salinity conditions are meant and not necessarily 'regime shifts' as "large, abrupt, persistent changes in the structure and function of a system". Thermal pollution or salt or freshwater discharge is definitely a pressure. Changes in thermal or salinity condition can also be effect of extractions, obstructions, global change, etc., and there with an impact. Clarify 'regime shift' as used here and elsewhere. In the questionnaire (shown in S1), this pressure was given as a 'regime change (thermal)' or 'regime change (salinity)'. In the main text, we have used these terms rather than 'regime shift', which has a more specific interpretation, as the reviewer describes. (The file with the supplementary material referred to is (still) named 'What is your gender'. – Please change the file-name. We have changed the file name to Painting et al_S1_Questionnaire_ P1-L19-L23: What is the difference between the main impacts and the key impacts; is it possible that those key impacts are actually effects of impacts? Clarify 'key impacts' as used here and elsewhere. The words 'main' and 'key' were used almost interchangeably. We have revised the manuscript to remove all use of 'key impacts'. P8-Ch3.1: Talking about

36 responses you mean '36 individuals who filled in form S1', that probably came with in total a huge amount of forms S2 and S3 I guess? How do you prevent getting a skewed/biased view on the theme as for instance almost 40% of the responses is from the UK? It seems that it is via working with 'national responses' as I gradually start to understand. Is there a pattern in the number of pressures or impacts identified per country with the number of responses per country than? To reduce/remove bias, we aggregated responses by country to give what we called a 'national view', i.e. a 'view by country'. We have edited the text in the Methodology (Section 2.1.1) to clarify this and added a sentence to clarify that 'These aggregated responses are referred to hereafter as 'national responses'. In Section 5.1, we explain that this was done to minimise bias. We analysed the data to determine if there was a pattern in the number of pressures or impacts identified per country vs the number of responses per country. The data indicate a weak relationship for both pressures and impacts. We have added a sentence on this to Section 3.1.

P8-L198: '100% of the national responses' – What does this mean? Are there responses considered not national (e.g. from researchers/people not working for the government)? Or does this mean that when one of the responses from a country includes the pressure (independent of the number of responses), it is considered to be identified for that country? This question is related to the reviewer's questions in the paragraph above. The responses were aggregated per country. So marine litter was identified by all responses at a national level but that does not mean it was identified in every single individual response from a country where there were multiple responses. This example has been included in Section 2.1.1. for clarity. There were two responses from people in organisations which represent multiple countries – Eurogoos, from a Swedish representative who answered from a Swedish perspective; and OSPAR from a UK-based person who answered for the region as a whole. This information is included in Supplementary material S3, Table S3.1. Supplementary Material (S2 and S3) was uploaded in the main manuscript during submission, but it appears that it may not have been included in the discussion paper. We have uploaded this material

separately from the main text in the revised manuscript. We have added text to this effect in Section 3.1. P13-Ch3.4.1: 'Responses to the questionnaire indicated that marine monitoring programmes provide less coverage of biological parameters than physical water column parameters and chemical parameters.' – Is this indeed the result per programme? Yes, Section 3.4 describes results per programme, as given by all respondents. Than the question arises about coverage (does monitoring take place at the same scale or with similar numbers of stations, or are there singular programmes covering large areas compared to several few station programmes for other aspects)? Is the presence or absence of monitoring of certain parameters at a national-subregional sea (or finer scale) level not a better indicator? Or are we only talking platform-based monitoring? Please discuss this? Section 3.5 describes the responses on the adequacy of monitoring programmes, and the explanations given for where monitoring was considered to be not adequate. Resolution in time and space was an important issue. These views were related to whole programmes, not only platform-based monitoring. The free text responses to the questionnaire did not go into in-depth analyses. These topics were expanded in the discussion, e.g. see Sections 5.4 and 5.5. P16-L358: Talking about percentages based on only 6 cases is a bit strange (17% is one respondent); at least do both. We have replaced the percentage with the number 'one', as suggested. P17-L372: 'Figure 133' should be 'Figure 13'. We have corrected the figure numbering, to be 'Figure 13'. P26-L642-643: 'Such monitoring programmes would require considerable effort, highlighting the need to define/characterize relevant environmental'. - What do you mean? Our apologies for the incomplete sentence. It should read 'Such monitoring programmes would require considerable effort, highlighting the need to define/characterize relevant environmental threats in each habitat or region'. This has now been corrected. Interactive comment on Ocean Sci. Discuss., https://doi.org/10.5194/os-2019-75, 2019.

Please also note the supplement to this comment:
https://www.ocean-sci-discuss.net/os-2019-75/os-2019-75-AC1-supplement.pdf

---

## Author Comment (AC2) · 2 Dec 2019

We would like to thank the reviewer for their comments and the opportunity to address these in the manuscript. Our responses to comments are shown below (in red text in our uploaded document) . We have tracked changes in the MSWord version of revised manuscript. These changes also address the comments of the other reviewer/s.

Reviewer 2: This paper presents the results of an opinion poll on marine monitoring needs and gaps based on a questionnaire filled by a number of scientists (36) from 12 European countries involved in the EU-project JERICO- Next. The topic of the questionnaire is interesting and timely. The paper is well written and perfectly fine as a

project deliverable, and as a way to advertise the project. To make it more interesting to a broad and specialised scientific community, a better balance should be taken among the topics stated in the abstract by giving more space to an updated synthesis of the literature and to an exhaustive and informative overview of the monitoring operations in place. As noted by the reviewer, this paper presents the results of the opinion poll. Very few respondents gave detail in the boxes for free text. The respondents did not supply an exhaustive overview of the monitoring operations in place. We have provided links to projects or websites where such overviews are underway. This approach would possibly allow to improve and sharpen the conclusions which are presently too general. As acknowledged by the Authors, the topic of the questionnaire has been addressed several times in recent years, but the most recent and relevant contributions to OceanObs '19 (published in Frontiers in Marine Science) are not considered. We have considered the most recent literature and improved and shortened our Conclusions by removing repetition and moving some of the text to Section 4, which has been edited to improve the synthesis of responses (these changes were extensive, and tracking was removed). The respondents had been given the task to act as national representatives, but it is not clear how this worked, because multiple questionnaires were filled for several countries, with 14 only from UK. This results in a lack of balance among countries and the bias that only countries and scientists participating in the project were directly involved, while it is hard to assess the actual coverage of those opinions not even in individual countries. To remove bias, we aggregated responses by country to give what we called a 'national view', i.e. a 'view by country'. We have edited the text in the Methodology (Section 2.1.1) to clarify this and added a sentence to clarify that 'These aggregated responses are referred to hereafter as 'national responses'. The other weakness is that the large part of the paper is based on opinions and partial overviews of the various issues. Although coming from qualified people and potentially interesting, it is hard to check their actual soundness and the completeness of the information. Some useful information is included in Supplementary material S3, Table S3.1, including the institutes that respondents belonged to. We

have added additional text on representatives in Section 3.1. With a broad range of opinions, and few or no detailed responses, we examined the issue which was raised by the majority of respondents, i.e. insufficient resolution in time and space. This is discussed in Sections 5.4 and 5.5. Among the improvements suggested in section 4, I was surprised not to find any references to advanced molecular technologies. We have broadened suggestions/recommendations for new technologies to include new methods such as molecular techniques. The most interesting part could have been the overview of the Monitoring Programmes in each country, but admittedly the reported programmes were a subsample rather than an exhaustive inventory. We acknowledge that the responses to the questionnaire do not represent a comprehensive overview so we are hesitant to draw too many conclusions based on information about a subset of European monitoring programmes. Nonetheless we have expanded this section to make the best of the responses we did get. The opinions of the respondents provided valuable information. A more recent, more quantitative study in Denmark (Andersen et al. 2019) using 35 databases yielded results which are broadly similar. We have referenced this study and other more recent studies in our revised manuscript.

Please also note the supplement to this comment:
https://www.ocean-sci-discuss.net/os-2019-75/os-2019-75-AC2-supplement.pdf